# Variability of the Brunt-Väisälä frequency at the OH*-layer height

Sabine Wüst [1], Michael Bittner [1, 2], Jeng-Hwa Yee [3] , Martin G. Mlynczak [4], James M. Russell III [5]

[1] Deutsches Fernerkundungsdatenzentrum, Deutsches Zentrum für Luft- und Raumfahrt, 82234 Oberpfaffenhofen, Germany

[2] Institut für Physik, Universität Augsburg, 86159 Augsburg, Germany

[3] Applied Physics Laboratory, The Johns Hopkins University, Laurel, Maryland, USA

[4] NASA Langley Research Center, Hampton, USA

[5] Center for Atmospheric Sciences, Hampton, USA

*Correspondence to*: Sabine Wüst (sabine.wuest@dlr.de)

**Abstract.** In and near the Alpine region, the most dense sub-network of identical NDMC instruments (Network for the Detection of Mesospheric Change, http://wdc.dlr.de/ndmc) can be found: five stations are equipped with OH*-spectrometers which deliver a time series of mesopause temperature each cloudless or only partially cloudy night. These measurements are suitable for the derivation of the density of gravity wave potential energy, provided that the Brunt-Väisälä frequency is known.

However, OH*-spectrometers do not deliver vertically-resolved temperature information, which is necessary for the calculation of the Brunt-Väisälä frequency. Co-located measurements or climatological values are needed.

We use 14 years of satellite-based temperature data (TIMED-SABER, 2002–2015) to investigate the inter- and intra-annual variability of the Brunt-Väisälä frequency at the OH*-layer height between 43.93–48.09°N and 5.71–12.95°E and provide a climatology.

# 1 Introduction

The Brunt-Väisälä frequency (BV frequency) is an important parameter in gravity wave theory. It is not only the highest possible frequency for gravity waves, it is also necessary when calculating different gravity wave parameters such as the density of wave potential energy averaged over a specific time period (see e.g. Wüst et al., 2016)

$$E_{pot}\left(\overline{T'^2}, T, \frac{dT}{dz}\right) = \frac{1}{2}\frac{g^2\overline{T'^2}}{N^2} \tag{1}$$

where

$N$ is the BV frequency,

$g$ the acceleration due to gravity,

$T$ the temperature, and

$\overline{T'^2}$ the mean squared normalized temperature fluctuation, i.e., the mean squared temperature fluctuation relative to the background temperature. It is calculated as follows:

$$\overline{T'^2} = \frac{1}{n}\sum_{i=1}^{n} {T'_i}^2 \tag{2}$$

with the normalized temperature fluctuation $T'_i$ at time step $i$ of $n$ time steps in total.

Energy and momentum are transported by gravity waves over large distances. Through interactions with other dynamical processes in the atmosphere (such as planetary waves, tides, other gravity waves) they can strongly influence atmospheric dynamics and are therefore regarded as an essential mechanism within atmospheric layer coupling. Case studies (e.g. Lu et al., 2015; Lu et al., 2009) based on LIDAR data (in the strato- and mesosphere) show that the amount of potential energy is

not constant with height and that the relation of potential and kinetic energy also varies height-dependently. Tsuda et al. (2000), for example, report that kinetic energy density dominates potential energy density (per unit mass) by a factor of 5/3 to 2 in the stratosphere based on GPS radio occultation data. Placke et al. (2013) use LIDAR data and show a minor deviation at mesopause heights from the value of 5/3 which is expected following linear gravity wave theory.

Also the BV frequency is not constant with height since it varies with the temperature and its vertical gradient as the following formula shows (e.g. Andrews, 2000):

$$N\left(T, \frac{dT}{dz}\right) = \sqrt{\frac{g}{T}\left(\frac{dT}{dz} - \Gamma_d\right)} \tag{3}$$

where $\Gamma_d$ is the dry-adiabatic lapse rate defined as the vertical adiabatic temperature decrease with a value of 9.8 K/km. This formula refers to the angular BV frequency. Even if not explicitly mentioned in the following, the terms BV frequency or BV period always denote the angular values.

Many measurement techniques suitable for the investigation of gravity waves provide vertical temperature profiles; this allows the direct calculation of the BV frequency and the density of wave potential energy (see e.g. Kramer et al., 2015; Mzé et al., 2014; Rauthe et al., 2008 to mention just a few). For OH* observation techniques, the situation is different: OH* spectrometers deliver information about temperature, also horizontally-resolved—if operated in a scanning mode (see e.g. Wachter et al., 2015)—but vertically averaged over the OH*-layer. OH* imaging systems provide brightness maps (e.g.

Sedlak et al. (2016) and Hannawald et al. (2016) who address a small part of the sky and Garcia et al. (1997) who operate an all-sky system), they do not provide temperature information for the majority of instruments. This is only possible when using narrow-band filters (see Pautet et al., 2014).

In order to deduce the density of wave potential energy from OH* spectrometer measurements, one needs to rely on

temperature climatologies or complementary measurements for the derivation of the BV frequency. While the latter might be of higher accuracy in most cases, lack of coincidence in either time or space of the complementary measurement with the passage of a wave could result in unrepresentative BV values (see Wendt et al., 2013 for the quantification of typical temperature differences due to mistime and misdistance).

In our preceding publication Wüst et al. (2016), we used TIMED-SABER (Thermosphere Ionosphere Mesosphere Energetics Dynamics, Sounding of the Atmosphere using Broadband Emission Radiometry) measurements for this purpose with the focus on three mid-European and one northern-European NDMC-station (Network for the Detection of Mesospheric Change, http://wdc.dlr.de/ndmc). The BV (angular) frequency derived for the OH*-layer height for the mid-latitude station Haute-Provence (43.93°N, 5.71°E, OHP), France is 0.022 s$^{-1}$ (yearly average) with a standard deviation of 0.002 s$^{-1}$ showing

a minimum in winter and a maximum in summer. With a yearly mean of 0.021 s$^{-1}$ and the same standard deviation as for OHP, it is slightly lower for the high-latitude station ALOMAR (69.28°N, 16.01°E), Norway. For measurements at the Urbana Atmospheric Observatory (40°N, 88°W), United States of America, over a 6-month period from January through June 1991, Bills and Gardner (1993) report a BV period $2\pi/N$ near 90 km height of 5.2 min ($\approx 0.020$ s$^{-1}$) during the winter months and 4.3 min ($\approx 0.024$ s$^{-1}$) during spring and early summer. She et al. (1991) compute a BV frequency of $2.12\times10^{-2}$ s$^{-1}$

($\approx 4.9$ min) and $2.29\times10^{-2}$ s$^{-1}$ ($\approx 4.6$ min) averaged between 86 km and 100 km for two nightly measurements in 1990 at Fort Collins (40.6°N, 105°W), United States of America.

Error propagation shows that an error of 10% in the BV frequency leads to an error of 20% in the density of wave potential energy $E_{pot}$:

$$\pm \left| \frac{\partial E_{pot}}{\partial N} \Delta N \right| \overset{(1)}{=} \pm \left| -2 \cdot \frac{g^2 \overline{T'^2}}{N^3} \cdot \Delta N \right| = \pm \left| \frac{g^2 \overline{T'^2}}{N^2} \cdot 2 \frac{\Delta N}{N} \right| = \pm \left| E_{pot} \cdot 2 \frac{\Delta N}{N} \right| \tag{4}$$

Since at least to our knowledge a temperature sounding satellite addressing the mesosphere is not planned for the time after
TIMED-SABER and in-situ measurements are rare and not available at every NDMC station, a climatology of the BV frequency is therefore very valuable for our purposes. Of course, gravity waves themselves influence the BV frequency, too. However, due to the thickness of the OH*-layer, small-scale variations cancel out (see e.g. Wüst et al., 2016). Furthermore, we plan to use this climatology for the calculation of the nightly-averaged gravity wave potential energy density based on NDMC measurements. So, the spatial averaging is accompanied by a temporal one which motivates the use of a climatology
in this case.

In the Alps and the vicinity of the Alps, there are five NDMC-stations: Oberpfaffenhofen (48.09°N, 11.28°E), the observatory Hohenpeißenberg (47.8°N, 11.0°E), the Environmental Research Station Schneefernerhaus (47.42°N, 10.98°E), Germany, and the observatories Haute Provence (43.93°N, 5.71°E), France, and Sonnblick (47.05°N, 12.95°E), Austria. This
is the most dense sub-network of NDMC stations. Therefore, we use vertical SABER profiles of the OH volume emission rate (VER, see section 2) in order to retrieve height and full width at half maximum (FWHM) of the OH*-layer for this geographical region. This information is necessary to calculate the BV frequency weighted for the OH*-layer (in the following denoted as OH*-equivalent BV frequency) based on vertical temperature profiles of SABER (section 3). We describe seasonal variations of the three parameters, height and FWHM of the OH*-layer as well as OH*-equivalent BV
frequency, discuss the results and provide a climatology of the yearly course of the OH*-equivalent BV frequency (section 4).

## 2 Data and analysis

### 2.1 Data

The TIMED satellite was launched on 7 December 2001 and the on-board limb-sounder SABER soon started to deliver
vertical profiles of kinetic temperature on a routine base from approximately 10 km to more than 100 km altitude with a
vertical resolution of about 2 km (Mertens et al., 2004; Mlynczak, 1997). The high vertical resolution is suitable for the
investigation of gravity wave activity. About 1200 temperature profiles are available per day. The latitudinal coverage on a
given day extends from about 52° latitude in one hemisphere to 83° in the other (Russell et al., 1999). Due to 180° yaw
maneuvers of the TIMED satellite this viewing geometry alternates once every 60 days (Russell et al., 1999). An overview
of the large number of SABER publications is available at http://saber.gats-inc.com/publications.php.

SABER temperatures are determined from measurements of infrared emission from carbon dioxide in the 15 $\mu$m spectral
interval. A comprehensive forward radiance model incorporating dozens of vibration-rotation bands of $CO_2$, including
isotopic and hot bands, and solving the full set of coupled radiative transfer equations under non-LTE, is the basis for the
SABER temperature retrievals. One of the main challenges in estimating kinetic temperature values from the $CO_2$ brightness
temperatures in the mesosphere and upper levels is certainly non-LTE conditions (NLTE), i.e., conditions that depart from
local thermodynamic equilibrium. NLTE algorithms for kinetic temperature were employed in the SABER temperature
retrieval from version 1.03 on (Lopez-Puertas et al., 2004; Mertens et al., 2004, 2008). Comparisons with reference data sets
generally confirm good quality of SABER temperatures (Remsberg et al., 2008).

We use TIMED-SABER temperature and OH-B channel data (volume emission rates, VER) in its latest version (2.0) for the
years 2002 to 2015. It was downloaded from the SABER homepage (saber.gats-inc.com). The OH-B channel covers the
wavelength range from 1.56 to 1.72 µm, which includes mostly the OH (4-2) and OH (5-3) vibrational transition bands. The
mean height difference of the OH (4-2)- and OH (3-1)-emission, which is addressed by the OH*-spectrometers at the Alpine
NDMC stations mentioned above, is approximately 500 m (von Savigny et al., 2012) and therefore negligible compared to
the FWHM.

According to Noll et al. (2016) and references therein, the total uncertainties for single temperature profiles are about 5 K at
90 km height including systematic uncertainties of ca. 3 K.

As mentioned above, we focus on NDMC stations in or near the Alps. Therefore, we use TIMED-SABER data between
43.93–48.09°N and 5.71–12.95°E. Since the OH*-spectrometers allow only measurements during night, we additionally
require SABER measurements between 17 UTC and 5 UTC.

## 2.2 Analysis

The squared BV frequency $N^2{}_i$ is calculated for each SABER height level $i$ between 70 and 100 km altitude where $m$ height levels exist in this height range. Information about the OH*-layer is determined from TIMED-SABER OH-VER profiles.

We obtain the maximum VER (in the following denoted as OH*-layer height) and the FWHM from the SABER data file which are then used for the calculation of the Gaussian-weighted squared BV frequency $\overline{N^2\left(T,\frac{dT}{dz}\right)}$. The assumption of a Gaussian-shaped OH*-layer is certainly simplified. In most cases, the OH*-layer follows a slightly asymmetric form with a positive skewness. That means the centroid height is a little bit higher (for example, ca. 0.7 km averaged over the first half of the year 2004) than the height of the maximum VER. Due to these small differences and the averaging which is applied

afterwards to the Gaussian-weighted squared BV frequency, this simplified approach can be justified.

In the following, the Gaussian-weighted squared BV frequency is referred to as squared OH*-equivalent BV frequency

$$\overline{N^2\left(T,\frac{dT}{dz}\right)} = \frac{1}{\sum_{i=1}^{m-1} f_i} \; \sum_{i=1}^{m-1} f_i \cdot \underbrace{\frac{2 \cdot g}{T_i + T_{i+1}}\left(\left(\frac{\Delta T}{\Delta z}\right)_i - \Gamma\right)}_{N^2{}_i} \tag{5}$$

where $\vec{f}$ is the vector of Gaussian weights for a mean equal to the maximum VER and a standard deviation σ as it is related

to the FWHM by

$$\text{FWHM} = 2\sqrt{2\ln 2}\,\sigma \; \approx 2.4\,\sigma \tag{6}$$

This was also the approach presented and discussed in Wüst et al. (2016) and Wüst et al. (2017).

It is also possible to first calculate the Gaussian-weighted temperature and its gradient over the OH*-layer and to use these

values afterwards for deriving a squared BV frequency $N^2\left(\bar{T},\frac{\overline{dT}}{dz}\right)$. The latter can be slightly different from $\overline{N^2\left(T,\frac{dT}{dz}\right)}$. Let us for example assume that $\frac{dT}{dz}$ is negative and constant, then height levels of lower temperature have a disproportionally higher BV frequency compared to height levels of higher temperature. Averaging over the whole height range leads to a lower BV frequency compared to the case of averaging temperature and its gradient first and calculating the BV frequency afterwards. Since gravity waves modulate the temperature, they also influence the BV frequency. If one calculates the BV

frequency for each height level separately and averages afterwards, as we do, one takes these gravity wave induced fluctuations into account but only according to the OH*-layer height and thickness.

With the selection criteria mentioned above, the number of data sets per year ranges between 509 (for the year 2002) and 590 (for the year 2011) although a matching profile is not available for every day.


## 3 Results and discussion

The yearly means of the three parameters, OH*-layer height, FWHM, and OH*-equivalent BV frequency, show nearly no variations for the years 2002–2015 (black line in fig. 1 (a)–(c)). They reach ca. 86.5 km and 7.5 km for the OH*-layer height and the FWHM, and 0.023 $s^{-1}$ for the OH*-equivalent BV frequency.

However, the yearly means of these parameters are accompanied by varying standard deviations (grey bars in fig. 1 (a)–(c)), which range between approximately 2% for the OH*-layer height, 10% for the OH*-equivalent Brunt-Väisälä frequency, and 20–30% for the FWHM. They are due to characteristic intra-annual variations of the parameters.

The yearly course of the OH*-layer height averaged over the years 2002–2015 varies between ca. 85 km and 87.5 km (thick line in fig. 2 (a)). The minimum is reached for the days of the year (DoY) 1–10 (January) and 320–366 (November–December), the maximum around DoY 90 and 220 (March–April and August). The mean FWHM has a minimum between 6 km and 6.5 km around DoY 180, and maxima at 9 km, 8 km and 8 km approximately for DoY 40 (February), 110 (April), and 285 (October) respectively (thick line in fig. 2 (b)). The mean OH*-equivalent BV frequency ranges between 0.021 $s^{-1}$ for DoY 40 (February) and 0.026 $s^{-1}$ for DoY 185 (July), approximately (fig. 2 (c), thick line). The coloured lines in fig. 2 (a)–(c) refer to the individual years and show 30-point running means.

So, one can say: the intra-annual variability dominates the inter-annual one by far. This provides the possibility to give an analytic description for the OH*-equivalent BV frequency which is identical for every year. The yearly course of the BV frequency is dominated by a minimum at the beginning of the year and a maximum in the middle of the year looking quite symmetric (see fig. 2 (c)), which suggests the use of a spectral analysis. Therefore, we apply harmonic analysis (all-step approach) to the daily mean data (diamonds in fig. 3). The harmonic analysis provides amplitude, phase, and period of the oscillations which explain the data variability best (see Bittner et al. (1994) or Wüst and Bittner (2006) for further information about the method). When searching for three oscillations, the annual, semi-annual, and ter-annual mode are found (information about amplitude and phases are given in table 1). The annual mode dominates the other two modes by a factor of 2–3. The semi- and the ter-annual mode are approximately of the same amplitude. The superposition of these three sinusoidals (solid line in fig. 3) explains ca. 74% of the data variability. The data deviate 16% at maximum from this curve; 84.4% and 97.8% of the data are located in a ±5%- and ±10%-interval (dashed lines in fig. 3) around the curve.

The superposition of only two oscillations explains 71% of the data variability. If one searches for four sinusoidals, an additional 60 d-oscillation with an amplitude of $0.02 \cdot 10^{-2}$ $s^{-1}$ is found. This is less than half of the amplitudes of the semi- and the ter-annual modes. This 60-day oscillation is probably not a geophysical period but may result instead from the local time sampling of the satellite or the fact that it performs a yaw maneuver once every 60 days (rotating through 180°) to keep

SABER viewing away from the sun. Therefore, we propose to use three sinusoidals with the parameters mentioned in table 1 to approximate the yearly course of the OH*-equivalent BV frequency with an uncertainty interval of ±5% or ±10% to include ca. 84% or 98% of the data.

As mentioned above, the total uncertainties for single SABER temperature profiles are about 5 K at 90 km height with systematic uncertainties of ca. 3 K. Since we calculate a mean of the OH*-equivalent BV frequency for every DoY using data of 14 years and approximate these values by a superposition of harmonic oscillations, we argue that we only have to pay attention to systematic uncertainties. If we assume that the systematic uncertainties change only slightly from one height step to the next one, then they mainly influence the absolute temperature value but not the temperature gradient. Let us assume a

"true" temperature $T_1 = 200\ K$ and a measured temperature $T_2 = 203\ K$, then the difference between the "true" squared BV frequency $N_1{}^2$ and the measured one $N_2{}^2$ relative to $N_1{}^2$ is:

$$\frac{N_1{}^2 - N_2{}^2}{N_1{}^2} = \frac{\frac{g}{T_1}\left(\frac{dT}{dz} - \Gamma\right) - \frac{g}{T_2}\left(\frac{dT}{dz} - \Gamma\right)}{\frac{g}{T_1}\left(\frac{dT}{dz} - \Gamma\right)} = \frac{\frac{1}{T_1} - \frac{1}{T_2}}{\frac{1}{T_1}} = \frac{T_2 - T_1}{T_2} \approx 1.5\% \tag{7}$$

For the (non-squared) BV frequency, the difference is ca. 0.75%. That means an uncertainty of ca. 3 K in temperature leads to a relative uncertainty of ca. 1.5% (0.75%) in the (non-)squared BV frequency. This is negligible when using the superposition of the annual, semi-annual, and ter-annual mode with an uncertainty interval of ± 5% or ±10% for the

approximation of the OH*-equivalent BV frequency.

A larger effect is caused by the height-dependence of $g$, which also influences $\Gamma$. According to Wüst et al. (2017), $g$ at mesopause height (in the following denoted with $g_{hd}$, hd for height-dependent) still reaches more than 97% compared to its surface value. Following CIRA-86, the temperature gradient at the mesopause ranges between ca. 1.4 and 2.9 K/km (see figure 4). The straight-forward calculation according to

$$\frac{N^2}{N_{hd}{}^2} = \frac{g}{g_{hd}} \cdot \frac{\frac{dT}{dz} - \frac{g}{c_p}}{\frac{dT}{dz} - \frac{g_{hd}}{c_p}} \tag{8}$$

shows that squared BV frequency $N_{hd}{}^2$ is ca. 7% lower compared to the case of constant $g$.

The dependence of the BV frequency on temperature and its vertical gradient causes the variability during the year. Due to the meridional circulation, the mesopause temperature is high in winter and low in summer. Since the inverse temperature is needed for the calculation of the BV frequency, the latter becomes low in winter and high in summer. This behaviour has

been reported previously by Bills and Gardner (1993) and Wüst et al. (2016). Figure 5 of Wüst et al. (2016) shows the OH*-equivalent BV frequency for the years 2012/13 above the station OHP based on TIMED-SABER and CIRA data. In contrast to the approach presented here, the OH*-height and its FWHM are kept constant (86.2 km and 7.9 km calculated for July

2012 to June 2013) there. Nevertheless, the SABER-based OH*-equivalent BV frequency is systematically higher than the one based on CIRA (0.019–0.022 s$^{-1}$) regardless of the calculation method employed here or in Wüst et al. (2016).

The OH*-emission height (and in some instances the FWHM of the OH*-layer) was already investigated on a case study
basis about 30–40 years ago mostly relying on rocket-borne or lidar measurements (Good, 1976; von Zahn et al., 1987; Baker and Stair, 1988). The investigation of the OH*-layer on a multi-year data basis started with the launch of WINDII (Wind Imaging Interferometer) on board of UARS (Upper Atmosphere Research Satellite) in September 1991. Due to the latitudinal range of 42° in one hemisphere to 72° in the other alternating every 36 days (e.g. Shepherd et al., 2006), publications using this data set like for example Zhang and Shepherd (1999) focus on the tropics and low mid-latitudes, and
are thus not suitable for a comparison with our results.

For SCIAMACHY (SCanning Imaging Absorption spectroMeter for Atmospheric CHartographY) on board of ENVISAT (ENVironmental SATellite), von Savigny (2015) published a mean OH (3-1) emission altitude for 40–50°N of 85.9 km (January 2003–December 2011, see his table 3). Due to the latitudinal coverage of SCIAMACHY, these values refer to September–March. For these months and the addressed latitudinal range (43.93–48.09°N), the emission altitude of the
SABER OH-B channel presented in our fig. 2 (a) (thick line) reaches 84.5–87.5 km and shows reasonable agreement with a mean value of ca. 86 km. In contrast to our analysis, von Savigny (2015) refers to the centroid altitude, while we show the altitude of maximum VER. These values differ, if the OH VER profile is asymmetric. Furthermore, remaining tidal effects due to different overpass times of both satellites and vertical shifts between the different Meinel-bands may also play a role. So, considering these possible sources of inconsistencies, the agreement is even quite good.

As stated by Shepherd et al. (2006), the OH excitation mechanism is driven by atomic oxygen which is produced at higher altitudes. All processes which lead to vertical transport of atomic oxygen rich or poor air from above or below influence the OH production: when atomic oxygen rich air is brought down, the VER increases but the peak emission height decreases and vice versa. This relationship can be used for inferring the OH*-height from ground-based measurements alone (Liu and Shepherd, 2006; Mulligan et al., 2009). Liu and Shepherd (2006) show that the OH VER profiles are also broadened when
the OH VER peak descends. This fits qualitatively to the yearly development of the FWHM and OH*-height (see fig. 2 (a) and (b)).

The latter seems to descend between 2002 and 2015. If one calculates the mean error of the yearly mean OH*-height which is the standard deviation (grey bars in fig. 2 (a)) divided by the square-root of the number of data points used per year (between 509 and 590, see section 2.2), the result reaches ca. 0.07 km (standard deviation of 1.5 km divided by $\sqrt{509}$). In
fig. 2 (a), the OH*-height descends ca. 0.25 km in 14 years (ca. 0.02 km/a) which is significant within the error bars. This value lies in the same range as the ones derived by Bremer and Peters (2008) for low-frequency reflection heights (ca. 80–83 km) and by Teiser and von Savigny (2017) for the OH(3-1) centroid altitude based on SCIAMACHY measurements. Unfortunately, the SCIAMACHY results refer to latitudes between 5°S and 30°N, higher northern latitudes are not covered. Bremer and Peters (2008) investigated the annual means of the low-frequency reflection heights measured at a constant solar

angle with mid-point of the transmission path at 50.71° N and 6.61° E between 1959 and 2006. After elimination of the solar- and geomagnetically-induced signal, the authors deduce a trend of -0.032 km/year. For the development of the OH*-equivalent BV frequency, this is currently not of importance: at least between 2007 and 2015, the OH*-equivalent BV frequency stayed constant (fig. 1 (c)). For the integration of a possible long-term development, it is therefore too early.

5   However, it shows that this question needs to be revisited in a couple of years.

**4 Summary and outlook**

We investigate the OH*-layer height, FWHM, and OH*-equivalent BV frequency based on 14 years of TIMED-SABER data for 43.93–48.09°N and 5.71–12.95°E.

5   Their annual means reach ca. 86.5 km, 7.5 km, and $2.35 \times 10^{-2}$ s$^{-1}$ and are nearly stable during 2002–2015. The characteristic intra-annual variations of the parameters lead to standard deviations of approximately 2% for the OH*-layer height, 10% for the OH*-equivalent BV frequency, and 20–30% for the FWHM.

Since the intra-annual variability dominates the inter-annual one by far, we can provide an analytic description for the mean OH*-equivalent BV frequency which is identical for every year. The superposition of an annual, semi-annual and ter-annual

10  oscillation explains ca. 74% of the data variability. Ca. 85% or 98% of the data are located in a ±5%- or ±10%-interval around the mean curve.

Similar investigations are planned for other NDMC stations in order to facilitate the estimation of the nightly mean density of wave potential energy independent of co-located measurements which deliver vertical temperature profiles.

**Acknowledgement**

The work of Sabine Wüst was funded by the Bavarian State Ministry for the Environment and Consumer Protection (VAO-project LUDWIG, project number TUS01 UFS-67093).

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

|  | "annual" oscillation | "semi-annual" oscillation | "ter-annual" oscillation |
| --- | --- | --- | --- |
| Period $T$ [d] | 364.7 | 182.4 | 121.3 |
| Amplitude $A$ [$10^{-2}$ s$^{-1}$] | 0.19 | 0.07 | 0.05 |
| Phase $\varphi$ [rad] | 1.70 | -1.69 | 2.36 |

**Table 1: Period, amplitude and phase of the three oscillations which explain the variability of the daily OH\*-equivalent BV frequency values (averaged over all years) best. They oscillate around a constant value of $2.32 \times 10^{-2}$ s$^{-1}$. The OH\*-equivalent BV frequency [s$^{-1}$] can be estimated by $2.32 \times 10^{-2} + \sum_{i=1}^{3} A_i \sin\left(\frac{2\pi}{T_i} \cdot DoY - \varphi_i\right)$. Due to leap years, the total amount of days for one year is set to 366, that means 1st March is DoY 61 for every year.**

(a)

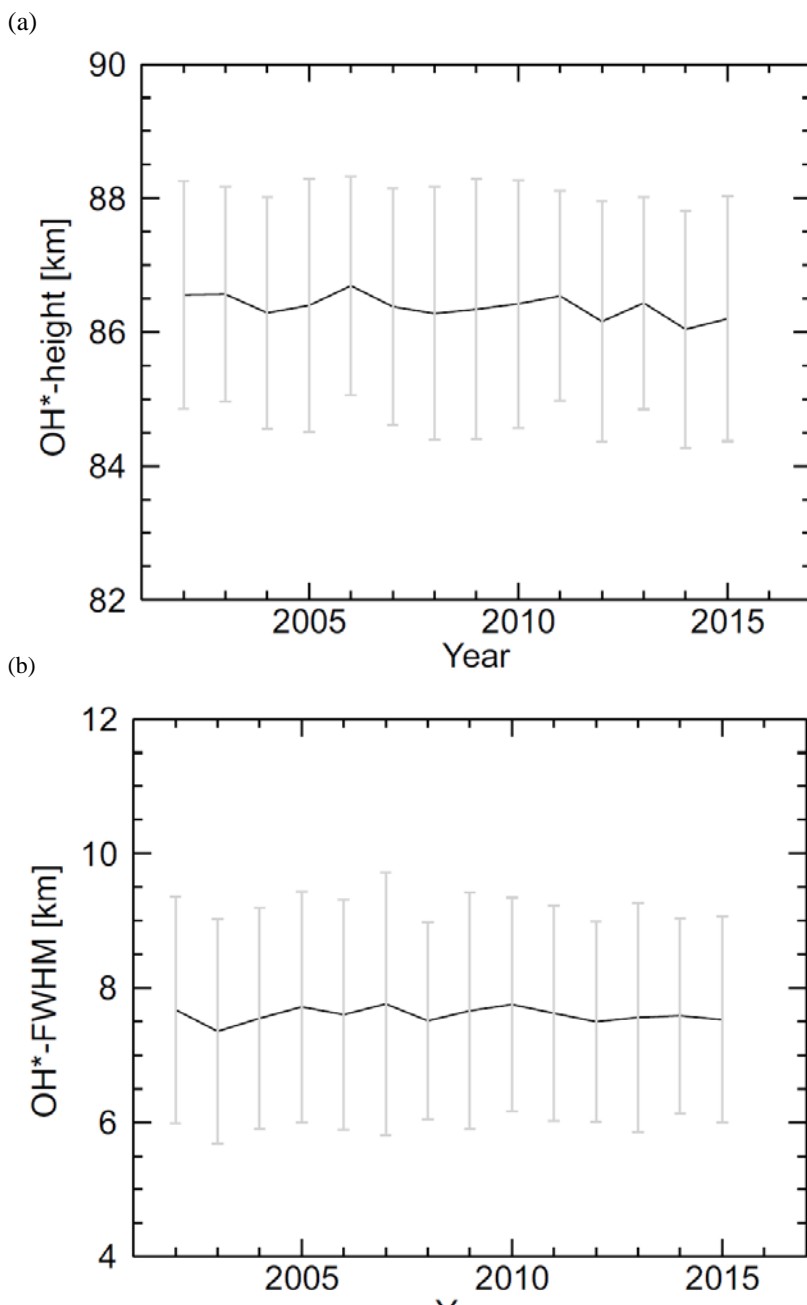

(b)

(c)

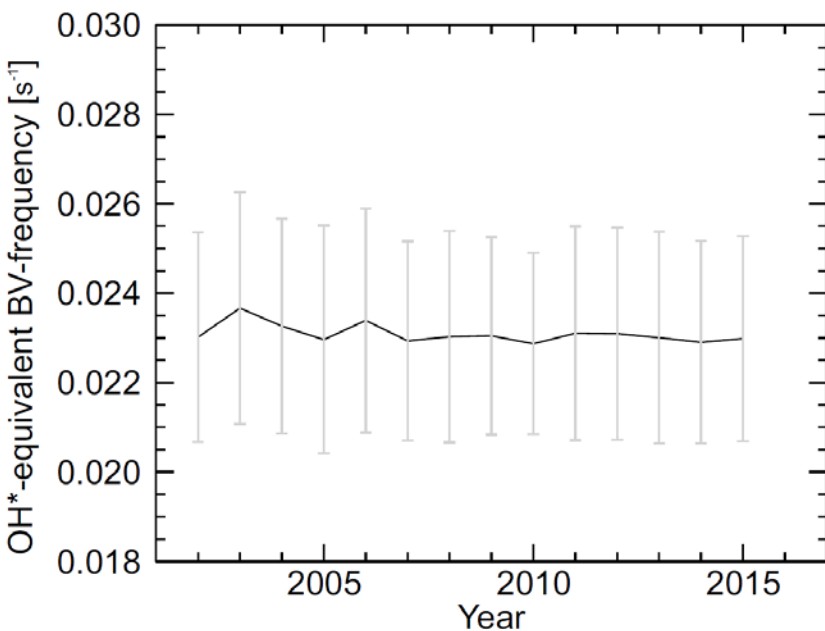

5  **Figure 1: The yearly means of the height of the maximal OH-VER (denoted as OH\*-height, see part a), the FWHM (part b), and the OH\*-equivalent BV frequency (part c) are nearly constant from 2002 to 2015 (black curve) but show comparatively large standard deviations (grey bars) for each year.**

(a)

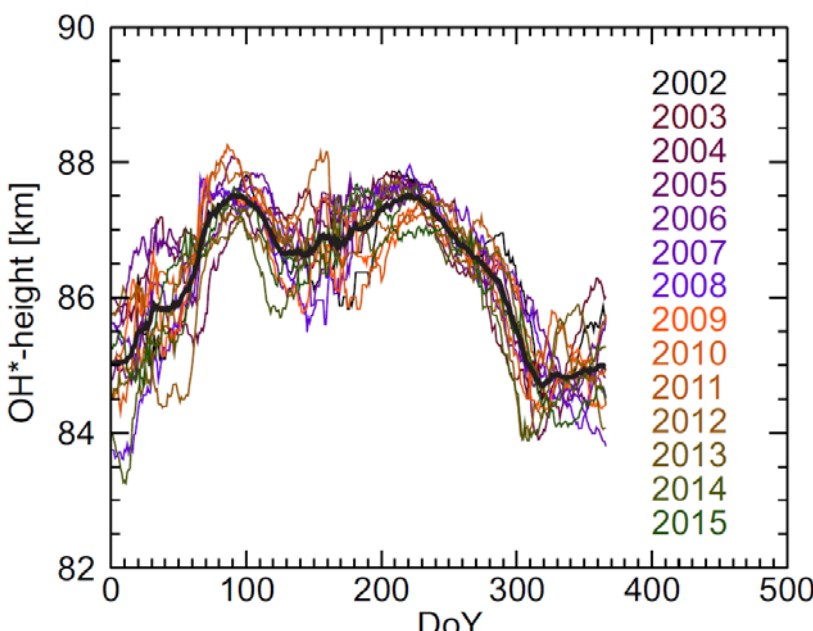

(b)

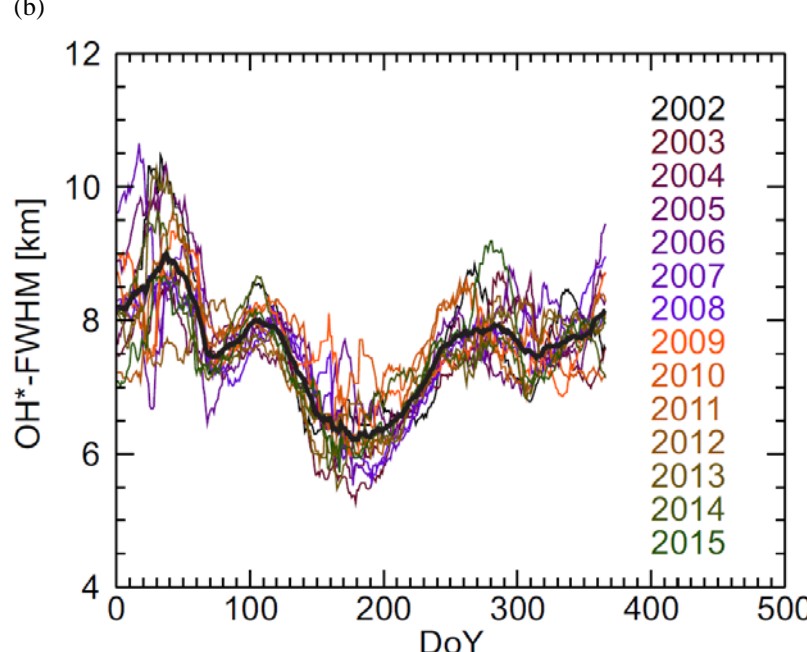

(c)

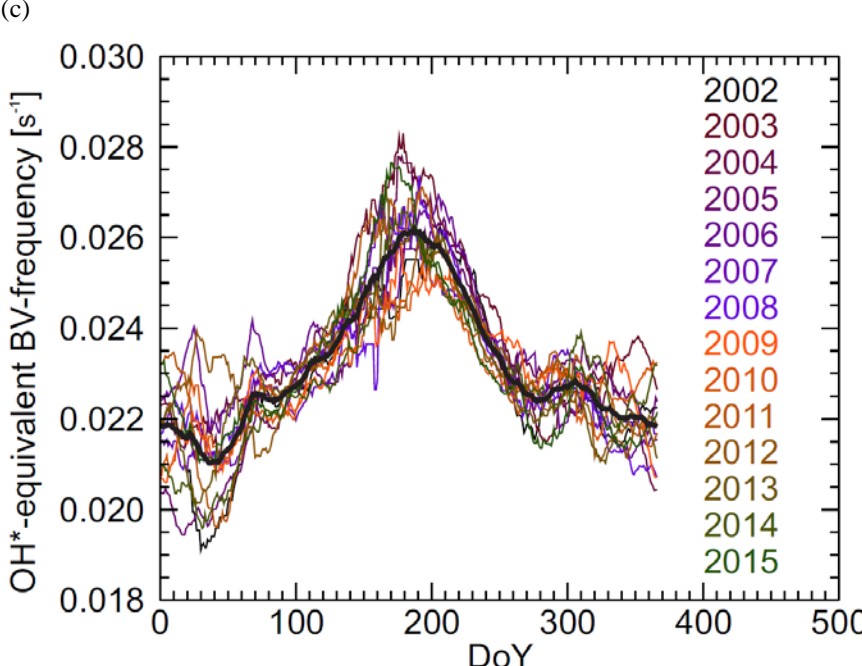

**Figure 2: OH\*-height (part a), FWHM (part b), and OH\*-equivalent BV frequency (part c) show characteristic variations during
the year (coloured lines: 30-point running means of the daily mean values with mirrored edge points for the beginning of 2002 and
the end of 2015). The values of the individual years deviate most from the mean over all years (black line) during winter and
especially at the beginning of the year. This might be due to enhanced atmospheric dynamics which is for example represented by
stratospheric warming events and its effects on the mesopause.**

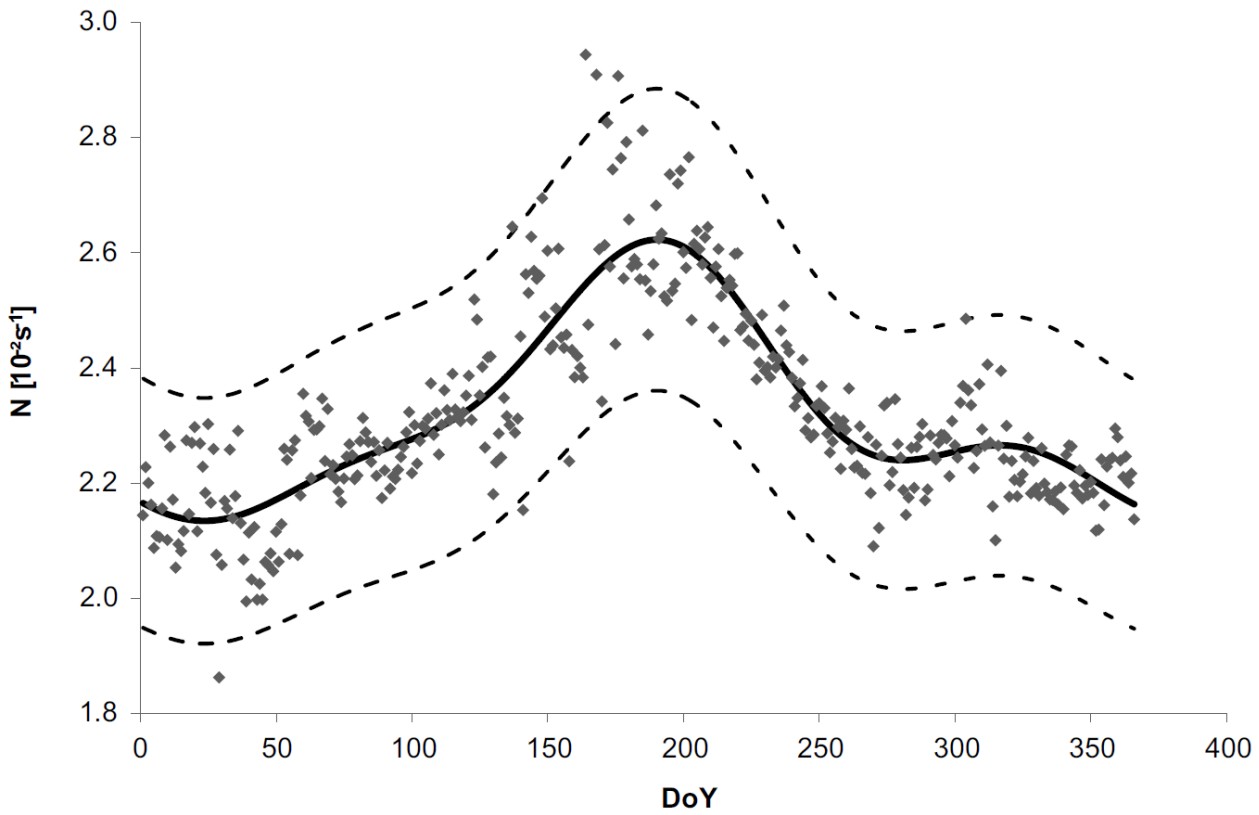

**Figure 3: The superposition of the annual, semi-, and ter-annual oscillation (solid line) explains ca. 74 % of the variability of the daily OH\*-equivalent BV frequency values averaged over all years (diamonds). 97.8% of the data are located in a ±10%-interval (dashed lines) around the curve.**

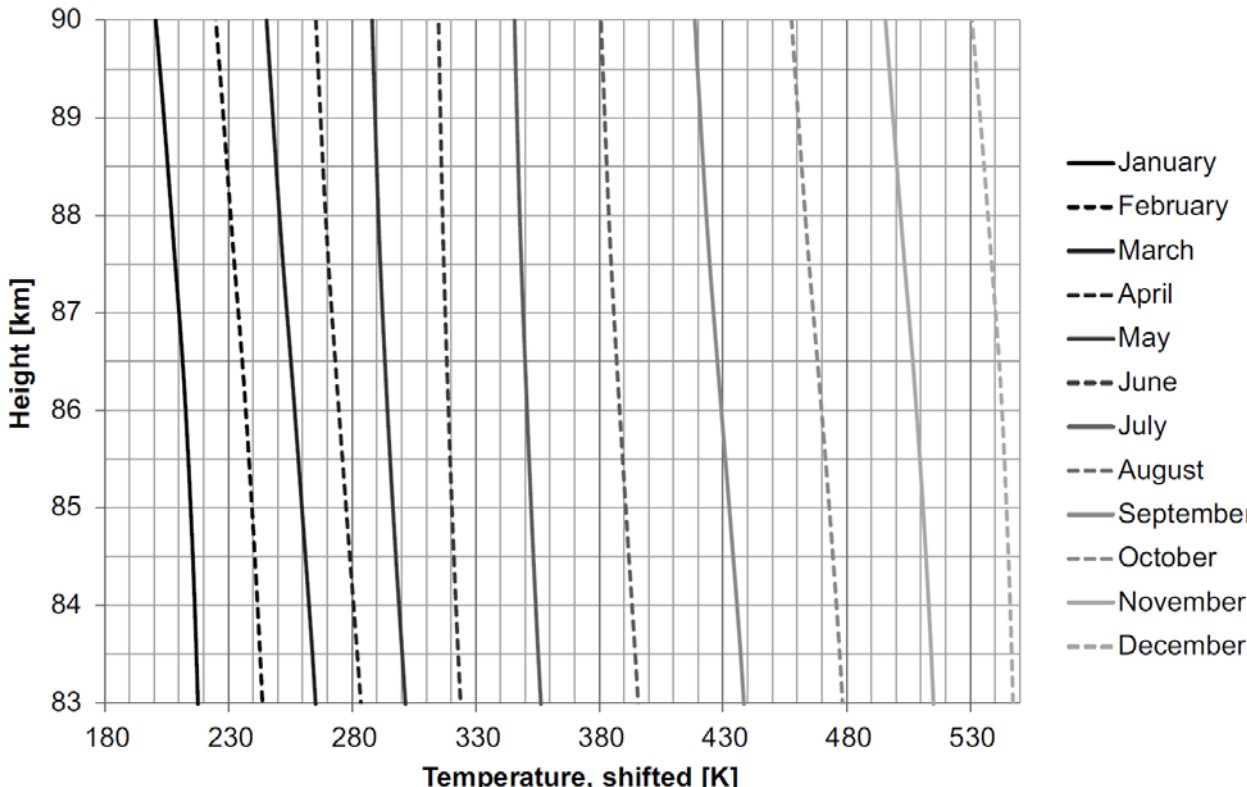

Figure 4: The temperature gradient based on CIRA-86 data for 45°N between 83 and 90 km height is negative during the whole year. The steepest gradient is reached in March and September (ca. 20 K / 7 km), it differs least from zero in summer (June–July, ca. 10 K / 7 km). The temperature values are offset by +30 K per month for all months except January.