# Peer review of "Variability of the Brunt-Väisälä frequency at the OH\*-layer height"

_Atmospheric Measurement Techniques, 2017_

## Referee Comment (RC1) · C. von Savigny (Referee) · 20 Jul 2017

General comments:

This is a generally well written study on the variability of the Brunt-Väisälä (BV) frequency in the MLT region. Knowledge of the BV frequency is relevant for the derivation of gravity wave related parameters, e.g. from ground-based observations of MLT temperature fluctuations. The results presented are useful for the aeronomy community and particularly for the groups operating ground-based OH rotational temperature spectrometers. I have no major objections against the publication of this manuscript, but ask the authors to consider the specific comments listed below.

Specific comments:

Page 1, line 16: "which are" -> "which is"

Page 2, line 25: "The same holds for the BV frequency"

It's not clear, what "The same" refers to. Please rephrase.

Page 3, line 25: I suggest mentioning the factor 2 pi in the context of BV period and BV frequency. I think the formula/values are not entirely consistent. Often the factor 2 pi is already included in the definition of the BV frequency. It should be clear, whether "frequency" refers to "angular frequency" or not.

Page 6, equation (4): I'm not sure the normalization by the norm of vector f is correct. One should divide by the sum of all elements of vector f, right? The norm, however, has a very different value, i.e. the square root of the summed up squared vector elements - at least according to the standard definition.

This probably only affects equation (4) and not the actual calculation of the OH* equivalent BV frequencies?

Page 6, line 5: Regarding the OH* layer height: If I understand correctly, the layer height is simply the height grid point with the maximum VER, right? It would be better to use centroid altitude, i.e. altitude weighted with the VER profile. If the altitude with maximum VER is used, the altitudes will be affected by the vertical sampling of the SABER limb measurements and by the retrieval altitude grid. I assume, the effects will be very small, though, but it would be good to motivate, why the height of the VER maximum is used here.

Also: the OH VER profile is not Gaussian. Assuming a Gaussian will also affect the results somewhat. I think you should at least mention that the actual VER profile is not Gaussian.

Page 9, line 11: "For ENVISAT [..] on board of SCIAMACHY" –> "For SCIAMACHY [..] on board of Envisat"

SCIAMACHY is the instrument, Envisat the satellite.

Page 9, line 15: Regarding the agreement between SCIA and SABER OH emission altitudes:

- Centroid altitude and altitude of maximum VER may be quite different (up to 2 km, I reckon), because the OH VER profile is asymmetric. Centroid altitude will be systematically larger than the VER-max altitude

- Remaining tidal effects between the average SABER local time and the SCIA local time (between 21 and 22 at 40 – 50 N) may also contribute to differences

- The vertical shifts between the different Meinel-bands may also play a role

So, considering these differences, the agreement is quite good.

Page 9, line 25/26: The linear trend in OH height is interesting and fairly consistent with a trend determined in our recent paper (Teiser & von Savigny, Variability of OH(3-1) and OH(6-2) emission altitude and volume emission rate from 2003 to 2011, JASTP, 161, 28-42, 2017). In this study, the trend in OH(3-1) centroid altitude (averaged between 5S and 30N) is about -20 m/yr. Higher northern latitudes are not covered, unfortunately. And one has to be careful, because trends in the SCIAMACHY limb pointing data may also play a role at this level. It is, however, interesting to note the qualitative and quantitative agreement between the different results.

References: The list of references contains several inconsistencies and typos, i.e.: spacing between initials is not consistent, e.g., "R. A." vs. "C.J."; in several cases the hyphen is missing between "Sol." and "Terr." for JASTP papers; in some cases there are periods between paper title and journal name, rather than commas.

Page 12, line 23: delete extra space in "T. ,"

Page 14, line 19: delete extra space in "OH (3-1)"

Page 14, line 2 bottom-up: delete extra space in "O (1S)"

Page 14, last line: comma after paper title missing.

---

## Referee Comment (RC2) · Anonymous Referee #2 · 24 Jul 2017

The authors describe a method of calculating a value for the Brunt-Väisälä (BV) frequency, that can be used at the altitude of OH* emissions near the mesopause (denoted OH*-equivalent BV frequency), based on temperature and volume emission rate (VER) profiles from the SABER instrument on the TIMED satellite.

They use 14 years of SABER profiles (2002-2015) in the vicinity of the Alpine region (43.93–48.09°N and 5.71–12.95°E) to obtain a climatology of the BV frequency in that region. They demonstrate that the BV frequency has an annual pattern which is repeated from year to year, even though there are considerable differences between individual years, with the largest variability occurring in the winter season. The climatology is specified in terms of an annual, semi-annual and ter-annual oscillations which account for 74% of the variation observed. Almost 98% of all of the nightly averaged

[Figure]

OH*-equivalent BV frequencies fall within the range of the climatology +/-10%.

The authors propose to use this climatology together with measurements of gravity waves obtained from a network of GRIPS-type (Ground-based Infrared P-branch Spectrometers) instruments already deployed in the Alpine region to enable them to estimate values of the nightly averaged density of potential energy (per unit mass) for the gravity waves detected. In an earlier publication, the authors reported that a 10% uncertainty in the BV frequency gives rise to a 20% uncertainty in the density of wave potential energy.

The manuscript is well organised and the intention of the authors is clear in almost all instances (however, see some of the specific comments below). The methods used to calculate the OH*-equivalent BV climatology are valid (see specific point relating to equation 4 on page 6) . The approach outlined could be employed by other ground-based observers, and it is therefore a valuable contribution to this field of study. The work is suitable for publication in AMT, provided that the specific points below are addressed.

Specific comments Page 1, line 14; rephrase 'the derivation of . . . Brunt-Väisälä frequency provided.' as 'the derivation of the density of gravity wave potential energy, provided that the Brunt-Väisälä frequency is known.' Page 2, line 3; replace 'like for example' by 'such as'. Page 2, line 8; g is the acceleration due to gravity, not the gravitational constant. Page 2, line 17; omit the word 'etc'. Page 2, line 20; the meaning of the phrase ' . . . nor the relation of potential and kinetic energy.' Is not clear. Please reword the entire sentence. Page 3, line 1; {uppercase greek gamma} (more usually written with a subscript-d) when referring to the dry adiabatic lapse rate) is defined as (gamma subscript-d = - dT/dz). Therefore the minus sign should be omitted and the phrase 'a value of' inserted before the numerical value. Page 3, line 4; suggest 'the direct calculation of' instead of 'to directly calculate'. Page 3, line 9; suggest 'do not provide temperature . . . ' instead of 'not even temperature . . .'. Page 3, lines 12-15; this sentence is unwieldy. It should be separated into two sentences. The first sen-

[Figure]

tence should end after 'the BV frequency' on line 13. The second sentence might be rephrased along the lines: 'While the latter might be of higher accuracy in most cases, lack of coincidence in either time or space of the complementary measurement with the passage of a wave could result in unrepresentative BV values'. Page 3, line 24; insert a comma after '(40°N, 88°W)'. Page 3, line 26; for clarity use '2.12×10-2 s-1' instead of '2.12Åů10-2 s-1' and use '($\sim$ 4.9 min)' instead of '(= 4.9 min)' on line 27. Page 4, line 12; replace 'denoted with' by 'denoted as'. Page 5, line 6; omit the word 'well' before 'suitable'. Page 5, lines 9/10; suggest rewording the sentence as follows: 'An overview of the large number of SABER publications is available at http://saber.gats-inc.com/publications.php.' Page 5, line 12; '15 $\mu$m' instead of '15 um'. Page 6, line 8; why does equation 4 contain 1/|f| instead of 1/(sum over i of fi) ?. Page 6, line 12; rephrase as 'This was also the approach presented and discussed in Wüst et al. (2016) and Wüst et al. (2017).'. Page 6, line 16; replace 'unproportionally' by 'disproportionally'. Page 6, line 24; replace 'whereas' by 'although'. Page 7, line 6; '0.023 s-1' would seem to be more accurate than '0.0235 s-1' for the OH*-equivalent BV value. Page 7, lines 14/15; suggest 'and maxima at 9 km, 8 km and 8 km approximately for DOY 40 (February), 110 (April), and 285 (October) respectively (thick line in fig. 2 (b)).' instead of 'and three maxima . . . and 285 (October, thick line in fig. 2 (b)).'. Page 7, line 21; replace 'mid' by 'middle'. Page 7, line 22; replace 'motivates' by 'suggests' and omit 'a' before 'harmonic'. Page 7, line 33; suggest replace the final two sentences by 'This 60-day oscillation is probably not a geophysical period but the may result instead from the local time sampling of the satellite or the fact that it performs a yaw maneuver once every 60 days (rotating through 180 degrees) to keep SABER viewing away from the sun.'. Page 8, line 2; The meaning of the sentence beginning 'Depending on the accuracy needed . . . ' is not clear. Please rephrase to clarify the intended point . Page 8, line 6; please be consistent in the use of 'DoY' or 'DOY' (lines 12-16 on page 7). Page 8, line 16; replace 'which influences also {uppercase greek gamma}.' by 'which also influences {uppercase greek gamma}.'. Page 8, lines 16-18; the sentence beginning 'According to Wüst et al. (2017) . . .' is confusing. It appears to confuse the variation

of g and {uppercase greek gamma} with altitude, and the effect of both of these on N-squared. The value and unit quoted on line 18 (9.81 K/km) as stated refer to g, but it is actually the unit of {uppercase greek gamma}. Please correct this sentence. Page 8, lines 24-25; suggest 'This behaviour has been reported previously by Bills and Gardner (1993) and Wüst et al. (2016).' instead of the sentence 'This behaviour . . . for example.' Page 8, line 27; suggest 'In contrast to the approach presented here . . . ' instead of 'Different to the approach presented here . . . '. Page 9, line 1; suggest 'Nevertheless, the SABER-based OH*-equivalent BV frequency is systematically higher than the one based on CIRA (0.019–0.022 1/s) regardless of the calculation method employed here or in Wüst et al. (2016).' instead of 'Independent of these facts, . . . CIRA (0.019–0.022 1/s)' Page 9, line 2; Please be consistent in the typography of units used for BV values (1/s) used here and also on page 17 and page 18 (y-axis label) or (s-1) used on pages 3, 7 and 10. Page 9, line 4; replace 'and in parts also' by 'in some instances'; replace 'on case study base' by 'on a case study basis'. Page 9, line 6; suggest replace 'base' by 'basis'. Page 9, lines 9-10; the emission altitude presented in Figure 2(a) is not the mean OH(3-1) emission altitude but is instead the emission altitude of the SABER OH-B channel as described on page 5 (lines 21-26). Figure 2(a) for the period September to March suggests that the mean emission altitude range is 85-87 km, not 86.0–86.5 km as stated. Page 9, lines 26 and 30; please use 'km/year' as the unit instead of 'km/a'. Page 10, line 5; use '2.35×10-2 s-1' instead of '2.35Âů10-2 s-1' and suggest 'during 2002-2015' instead of 'during 14 years'. Page 10, line 8; consider inserting the word 'mean' before 'OH*-'. Page 10, line 11; consider inserting the word 'mean' before 'curve'.

Page 12, lines 32-33 and page 13, lines 1-2; these references are not in alphabetical order of surname. Page 15, Table 1; Why not use column headings "annual", "semi-annual" and "ter-annual" (using quotation marks around each heading to indicate that they are approximate periods) instead of '1st oscillation 2nd oscillation 3rd oscillation'? Figure 3 on page 20 uses 'annual, semi-, and ter-annual' to describe these oscillations. Page 15, lines 4-5; use '2.32×10-2 s-1' instead of '2.32Âů10-2 s-1' Page 17, caption

of Figure 1; omit 'a' in '. . . but show a comparatively . . ..'.  Pages 18/19/20, label on x-axis is written as 'DoY', whereas it is written as 'DOY' on page 7 (lines12 – 16) and page 8 (line 6) as 'DoY'. Please be consistent in this label. Page 21, caption of Figure 4, final sentence; suggest 'The temperature values are offset by +30 K per month for all months except January.'.

Please also note the supplement to this comment:
https://www.atmos-meas-tech-discuss.net/amt-2017-191/amt-2017-191-RC2-supplement.pdf

---

## Author Comment (AC1) · 26 Sep 2017

Dear Christian,

Thank you very much for your valuable comments. I tried to include them as best as possible.

One thing I realized during the revision of the manuscript: On page 3, line 30 (original manuscript), I wrote "Error propagation shows that an error of 10% in the BV frequency leads to an error of 20% in the density of wave potential energy (see Wüst et al., 2016)." This mentioned calculation was included in the first version of Wüst et al. (2016). Due to re-arrangements of the manuscript in the review process, I deleted

it. Therefore, I now included the calculation in this manuscript and deleted the reference to Wüst et al. (2016).

General comments:

This is a generally well written study on the variability of the Brunt-Väisälä (BV) frequency in the MLT region. Knowledge of the BV frequency is relevant for the derivation of gravity wave related parameters, e.g. from ground-based observations of MLT temperature fluctuations. The results presented are useful for the aeronomy community and particularly for the groups operating ground-based OH rotational temperature spectrometers. I have no major objections against the publication of this manuscript, but ask the authors to consider the specific comments listed below.

Specific comments:

- Page 1, line 16: "which are" -> "which is" Done

- Page 2, line 25: "The same holds for the BV frequency" It's not clear, what "The same" refers to. Please rephrase. Done

- Page 3, line 25: I suggest mentioning the factor 2 pi in the context of BV period and BV frequency. I think the formula/values are not entirely consistent. Often the factor 2 pi is already included in the definition of the BV frequency. It should be clear, whether "frequency" refers to "angular frequency" or not.

  I inserted the following sentences after formula (3) "This formula refers to the angular BV frequency. Even if not explicitly mentioned in the following, the terms BV frequency or BV period always denote the angular values." Furthermore, I included $2\pi/N$ after BV period (former page 3, line 25).

- Page 6, equation (4): I'm not sure the normalization by the norm of vector f is correct. One should divide by the sum of all elements of vector f, right? The

norm, however, has a very different value, i.e. the square root of the summed up squared vector elements - at least according to the standard definition.

You are right, the calculation is correct but the formula is wrong. I corrected it.

This probably only affects equation (4) and not the actual calculation of the OH* equivalent BV frequencies?

- Page 6, line 5: Regarding the OH* layer height: If I understand correctly, the layer height is simply the height grid point with the maximum VER, right? Yes, that's true

It would be better to use centroid altitude, i.e. altitude weighted with the VER profile. If the altitude with maximum VER is used, the altitudes will be affected by the vertical sampling of the SABER limb measurements and by the retrieval altitude grid. I assume, the effects will be very small, though, but it would be good to motivate, why the height of the VER maximum is used here.

I analysed the first half of the year 2004. This year was arbitrarily chosen. The mean difference between the centroid altitude and the peak altitude is ca. 0.7 km, the skewness of the VER-distribution is 0.8 which is not a very large value.

Since the vertical resolution of the SABER data is ca. 300–400 m, a difference of 0.7 km corresponds to 2 data points at maximum. Taking into account the FWHM of 7–8 km of the OH*-layer and the calculation method of the climatology of the Brunt-Väisälä frequency (least squares fit to the daily mean values of the Brunt-Väisälä frequency), I would judge the effect as negligible.

I inserted in the manuscript: "The assumption of a Gaussian-shaped OH*-layer is certainly simplified. In most cases, the OH*-layer follows a slightly asymmetric form with a positive skewness. That means the centroid height is a little bit higher (for example, ca. 0.7 km averaged over the first half of the

year 2004) than the height of the maximum VER. Due to these small differences and the averaging which is applied afterwards to the Gaussian-weighted squared BV frequency, this simplified approach can be justified."

Also: the OH VER profile is not Gaussian. Assuming a Gaussian will also affect the results somewhat. I think you should at least mention that the actual VER profile is not Gaussian.

See above.

- Page 9, line 11: "For ENVISAT [..] on board of SCIAMACHY" –> "For SCIAMACHY [..] on board of Envisat" SCIAMACHY is the instrument, Envisat the satellite. Done.

- Page 9, line 15: Regarding the agreement between SCIA and SABER OH emission altitudes:

Centroid altitude and altitude of maximum VER may be quite different (up to 2 km, I reckon), because the OH VER profile is asymmetric. Centroid altitude will be systematically larger than the VER-max altitude

Remaining tidal effects between the average SABER local time and the SCIA local time (between 21 and 22 at 40 – 50 N) may also contribute to differences

The vertical shifts between the different Meinel-bands may also play a role So, considering these differences, the agreement is quite good.

Thank you for this hint. I mentioned it in the manuscript "In contrast to our analysis, von Savigny (2015) refers to the centroid altitude, while we show the altitude of maximum VER. These values differ, if the OH VER profile is asymmetric. Furthermore, remaining tidal effects due to different overpass times of both satellites and vertical shifts between the different Meinel-bands may also play a role. So, considering these possible sources of inconsistencies, the agreement is even quite good."

[Figure]

- Page 9, line 25/26: The linear trend in OH height is interesting and fairly consistent with a trend determined in our recent paper (Teiser & von Savigny, Variability of OH(3-1) and OH(6-2) emission altitude and volume emission rate from 2003 to 2011, JASTP, 161, 28-42, 2017). In this study, the trend in OH(3-1) centroid altitude (averaged between 5S and 30N) is about -20 m/yr. Higher northern latitudes are not covered, unfortunately. And one has to be careful, because trends in the SCIAMACHY limb pointing data may also play a role at this level. It is, however, interesting to note the qualitative and quantitative agreement between the different results.

  Indeed, that's interesting and I included it therefore in the manuscript p. 12, ll.6–8 (version with changes marked).

- References: The list of references contains several inconsistencies and typos, i.e.: spacing between initials is not consistent, e.g., "R. A." vs. "C.J."; in several cases the hyphen is missing between "Sol." and "Terr." for JASTP papers; in some cases there are periods between paper title and journal name, rather than commas.

  Page 12, line 23: delete extra space in "T. ," Done.

  Page 14, line 19: delete extra space in "OH (3-1)" Done.

  Page 14, line 2 bottom-up: delete extra space in "O (1S)" Done.

  Page 14, last line: comma after paper title missing. Done.

  I checked the whole reference list for inconsistencies and hope that I could identify all.

---

## Author Comment (AC2)

Thank you very much for your valuable comments. I tried to include them as best as possible.

One thing I realized during the revision of the manuscript: On page 3, line 30 (original manuscript), I wrote "Error propagation shows that an error of 10% in the BV frequency leads to an error of 20% in the density of wave potential energy (see Wüst et al., 2016)." This mentioned calculation was included in the first version of Wüst et al. (2016). Due to re-arrangements of the manuscript in the review process, I deleted it. Therefore, I now included the calculation in this manuscript and deleted the reference to Wüst et al. (2016).

[Figure]

The authors describe a method of calculating a value for the Brunt-Väisälä (BV) frequency, that can be used at the altitude of OH\* emissions near the mesopause (denoted OH\*-equivalent BV frequency), based on temperature and volume emission rate (VER) profiles from the SABER instrument on the TIMED satellite.

They use 14 years of SABER profiles (2002-2015) in the vicinity of the Alpine region (43.93–48.09°N and 5.71–12.95°E) to obtain a climatology of the BV frequency in that region. They demonstrate that the BV frequency has an annual pattern which is repeated from year to year, even though there are considerable differences between individual years, with the largest variability occurring in the winter season. The climatology is specified in terms of an annual, semi-annual and ter-annual oscillations which account for 74% of the variation observed. Almost 98% of all of the nightly averaged OH\*-equivalent BV frequencies fall within the range of the climatology +/-10%.

The authors propose to use this climatology together with measurements of gravity waves obtained from a network of GRIPS-type (Ground-based Infrared P-branch Spectrometers) instruments already deployed in the Alpine region to enable them to estimate values of the nightly averaged density of potential energy (per unit mass) for the gravity waves detected. In an earlier publication, the authors reported that a 10% uncertainty in the BV frequency gives rise to a 20% uncertainty in the density of wave potential energy.

The manuscript is well organised and the intention of the authors is clear in almost all instances (however, see some of the specific comments below). The methods used to calculate the OH\*-equivalent BV climatology are valid (see specific point relating to equation 4 on page 6) . The approach outlined could be employed by other ground-based observers, and it is therefore a valuable contribution to this field of study. The work is suitable for publication in AMT, provided that the specific points below are addressed.

[Figure]

Specific comments

Page 1, line 14; rephrase 'the derivation of … Brunt-Väisälä frequency provided.'
as 'the derivation of the density of gravity wave potential energy, provided that the
Brunt-Väisälä frequency is known.' Done

Page 2, line 3; replace 'like for example' by 'such as'. Done

Page 2, line 8; g is the acceleration due to gravity, not the gravitational constant.
Done

Page 2, line 17; omit the word 'etc'. Done

Page 2, line 20; the meaning of the phrase ' … nor the relation of potential and
kinetic energy.' Is not clear. Please reword the entire sentence. Done

Page 3, line 1; {uppercase greek gamma} (more usually written with a subscript-d)
when referring to the dry adiabatic lapse rate) is defined as (gamma subscript-d = -
dT/dz). I additionally provided this information to avoid confusion.

Therefore the minus sign should be omitted and the phrase 'a value of' inserted
before the numerical value. I wrote "where $\Gamma_d$ is the dry-adiabatic lapse rate defined
as the vertical adiabatic temperature decrease with a value of 9.8 K/km."

Page 3, line 4; suggest 'the direct calculation of' instead of 'to directly calculate'.
Done

Page 3, line 9; suggest 'do not provide temperature … ' instead of 'not even
temperature …'. Done

Page 3, lines 12-15; this sentence is unwieldy. It should be separated into two
sentences. The first sentence should end after 'the BV frequency' on line 13. The
second sentence might be rephrased along the lines: 'While the latter might be of

higher accuracy in most cases, lack of coincidence in either time or space of the complementary measurement with the passage of a wave could result in unrepresentative BV values'. Done

Page 3, line 24;insert a comma after '(40°N, 88°W)'. Done

Page 3, line 26; for clarity use '2.12×10-2 s-1' instead of '2.12Åů10-2 s-1' and use '(~

4.9 min)' instead of '(= 4.9 min)' on line 27. Done, also for $2.29×10^{-2}$ in the same line. "≈" also inserted in the lines above.

Page 4, line 12; replace 'denoted with' by 'denoted as'. Done

Page 5, line 6; omit the word 'well' before 'suitable'. Done

Page 5, lines 9/10; suggest rewording the sentence as follows: 'An overview of the large number of SABER publications is available at http://saber.gats-inc.com/publications.php.' Done

Page 5, line 12; '15 $\mu$m' instead of '15 um'. Done

Page 6, line 8; why does equation 4 contain 1/|f| instead of 1/(sum over i of fi) ?. Corrected, the calculation is right, the formula was wrong.

Page 6, line 12; rephrase as 'This was also the approach presented and discussed in Wüst et al. (2016) and Wüst et al. (2017).'. Done

Page 6, line 16; replace 'unproportionally' by 'disproportionally'. Done

Page 6, line 24; replace 'whereas' by 'although'. Done

Page 7, line 6; '0.023 s-1' would seem to be more accurate than '0.0235 s-1' for the OH*-equivalent BV value. Ok

Page 7, lines 14/15; suggest 'and maxima at 9 km, 8 km and 8 km approximately for DOY 40 (February), 110 (April), and 285 (October) respectively (thick line in fig. 2 (b)).' instead of 'and three maxima . . . and 285 (October, thick line in fig. 2 (b)).'.

[Figure]

Done

Page 7, line 21; replace 'mid' by 'middle'. Done

Page 7, line 22; replace 'motivates' by 'suggests' and omit 'a' before 'harmonic'. Done

Page 7, line 33; suggest replace the final two sentences by 'This 60-day oscillation is probably not a geophysical period but the may result instead from the local time sampling of the satellite or the fact that it performs a yaw maneuver once every 60 days (rotating through 180 degrees) to keep SABER viewing away from the sun.'. Done but left out "the" before "may result".

Page 8, line 2; The meaning of the sentence beginning 'Depending on the accuracy needed … ' is not clear. Please rephrase to clarify the intended point . Sentence changed accordingly.

Page 8, line 6; please be consistent in the use of 'DoY' or 'DOY' (lines 12-16 on page 7). Done, changed to DoY everywhere in the document.

Page 8, line 16; replace 'which influences also {uppercase greek gamma}.' by 'which also influences {uppercase greek gamma}.'. Done

Page 8, lines 16-18; the sentence beginning 'According to Wüst et al. (2017) …' is confusing. It appears to confuse the variation of g and {uppercase greek gamma} with altitude, and the effect of both of these on N-squared. The value and unit quoted on line 18 (9.81 K/km) as stated refer to g, but it is actually the unit of {uppercase greek gamma}. Please correct this sentence. Done

Page 8, lines 24-25; suggest 'This behaviour has been reported previously by Bills and Gardner (1993) and Wüst et al. (2016).' instead of the sentence 'This behaviour … for example.' Done

Page 8, line 27; suggest 'In contrast to the approach presented here … ' instead of 'Different to the approach presented here … '. Done

[Figure]

Page 9, line 1; suggest 'Nevertheless, the SABER-based OH\*-equivalent BV frequency is systematically higher than the one based on CIRA (0.019–0.022 1/s) regardless of the calculation method employed here or in Wüst et al. (2016).' instead of 'Independent of these facts, … CIRA (0.019–0.022 1/s)' Done

Page 9, line 2; Please be consistent in the typography of units used for BV values (1/s) used here and also on page 17 and page 18 (y-axis label) or (s-1) used on pages 3, 7 and 10. Changed to s-1.

Page 9, line 4; replace 'and in parts also' by 'in some instances'; replace 'on case study base' by 'on a case study basis'. Done

Page 9, line 6; suggest replace 'base' by 'basis'. Done

Page 9, lines9-10; the emission altitude presented in Figure 2(a) is not the mean OH(3-1) emission altitude but is instead the emission altitude of the SABER OH- B channel as described on page 5 (lines 21-26). Figure 2(a) for the period September to March suggests that the mean emission altitude range is 85-87 km, not 86.0–86.5 km as stated. This comment presumably refers to lines 14–15. The formulation I used was misleading, I am sorry for that. I changed the sentences now to "For these months and the addressed latitudinal range (43.93–48.09°N), the emission altitude of the SABER OH-B channel presented in our fig. 2 (a) (thick line) reaches 84.5-87.5 km and shows reasonable agreement with a mean value of ca. 86 km."

Page 9, lines 26 and 30; please use 'km/year' as the unit instead of 'km/a'. Done

Page 10, line 5; use '2.35×10-2 s-1' instead of '2.35Åu˚ 10-2 s-1' and suggest 'during 2002-2015' instead of 'during 14 years'. Done

Page 10, line 8; consider inserting the word 'mean' before 'OH\*-'. Done

Page 10, line 11; consider inserting the word 'mean' before 'curve'. Done

Page 12, lines 32-33 and page 13, lines 1-2; these references are not in alphabetical order of surname. Done

[Figure]

Page 15, Table 1; Why not use column headings "annual", "semi- annual" and "ter-annual" (using quotation marks around each heading to indicate that they are approximate periods) instead of '1st oscillation 2nd oscillation 3rd oscillation'?

Figure 3 on page 20 uses 'annual, semi-, and ter-annual' to describe these oscillations. Changed

Page 15, lines 4-5; use '2.32×10-2 s-1' instead of '2.32Âu˚ 10-2 s-1' Done

Page 17, caption of Figure 1; omit 'a' in '. . . but show a comparatively . . ..'. Done

Pages 18/19/20, label on x-axis is written as 'DoY', whereas it is written as 'DOY' on page 7 (lines12 – 16) and page 8 (line 6) as 'DoY'. Please be consistent in this label. Changed DOY to DoY in the whole manuscript

Page 21, caption of Figure 4, final sentence; suggest 'The temperature values are offset by +30 K per month for all months except January.'. Done